# Estimation of the Grüneisen Parameter of High-Entropy Alloy-Type Functional Materials: The Cases of $REO_{0.7}F_{0.3}BiS_2$ and MTe

Fysol Ibna Abbas [1,2], Yuki Nakahira [1], Aichi Yamashita [1], Md. Riad Kasem [1], Miku Yoshida [1], Yosuke Goto [1], Akira Miura [3], Kensei Terashima [4], Ryo Matsumoto [4,5], Yoshihiko Takano [4], Chikako Moriyoshi [6] and Yoshikazu Mizuguchi [1,*]

1 Department of Physics, Tokyo Metropolitan University, Hachioji 192-0397, Japan; fysolibnaabbas@gmail.com (F.I.A.); yuki-nakahira@tmu.ac.jp (Y.N.); aichi@tmu.ac.jp (A.Y.); riad.phy.du@gmail.com (M.R.K.); yoshida-miku2@ed.tmu.ac.jp (M.Y.); y_goto@tmu.ac.jp (Y.G.)
2 Department of Electrical and Electronic Engineering, City University, Dhaka 1216, Bangladesh
3 Department of Applied Chemistry, Faculty of Engineering, Hokkaido University, Sapporo 060-8628, Japan; amiura@eng.hokudai.ac.jp
4 International Center for Materials Nanoarchitectonics (MANA), National Institute for Materials Science, Tsukuba 305-0047, Japan; terashima.kensei@nims.go.jp (K.T.); matsumoto.ryo@nims.go.jp (R.M.); takano.yoshihiko@nims.go.jp (Y.T.)
5 International Center for Young Scientists (ICYS), National Institute for Materials Science, Tsukuba 305-0047, Japan
6 Graduate School of Advanced Science and Engineering, Hiroshima University, Higashihiroshima 739-8526, Japan; moriyosi@sci.hiroshima-u.ac.jp
* Correspondence: mizugu@tmu.ac.jp

**Abstract:** In functional materials such as thermoelectric materials and superconductors, the interplay between functionality, electronic structure, and phonon characteristics is one of the key factors to improve functionality and to understand the underlying mechanisms. In the first part of this article, we briefly review investigations on lattice anharmonicity in functional materials on the basis of the Grüneisen parameter ($\gamma_G$). We show that $\gamma_G$ can be a good index for large lattice anharmonicity and for detecting a change in anharmonicity amplitude in functional materials. Then, we show original results on the estimation of $\gamma_G$ for recently developed high-entropy alloy-type (HEA-type) functional materials with a layered structure and a NaCl-type structure. As a common trend for those two systems with two- and three-dimensional structures, we found that $\gamma_G$ increased with a slight increase in the configurational entropy of mixing ($\Delta S_{mix}$) and then decreased with increasing $\Delta S_{mix}$ in the high-entropy region.

**Keywords:** Grüneisen parameter; lattice anharmonicity; high-entropy alloy; superconductors; thermoelectric materials

## 1. Introduction

### 1.1. Thermoelectric Materials and Superconductors

In the last two decades, the development of effective energy-creating and energy-saving technologies has supplied interesting tools to solve the energy problem and to prevent climate change. One of the solutions is represented by thermoelectric (TE) materials and modules [1,2]. TE modules allow the direct conversion of unused thermal energy into useful electrical power, which can help to reduce carbon dioxide emissions and contribute to a more sustainable society. However, the performance of TE devices needs to be further improved for their practical application, and thus, the development of new TE materials that can allow us to fabricate high-performance TE devices is needed. To estimate the performance of TE materials, dimensionless figure-of-merit ($ZT$), which is calculated by the following Formula (1), is essential.

$$ZT = \frac{S^2 \sigma T}{\kappa} = \frac{S^2 \sigma T}{\kappa_{\mathrm{el}} + \kappa_{\mathrm{ph}}} \tag{1}$$

where $S$, $\sigma$, $\kappa$, and $T$ are Seebeck coefficient, electrical conductivity, thermal conductivity, and absolute temperature, respectively. Generally, $\kappa$ is considered as the sum of $\kappa_{\mathrm{el}}$ and $\kappa_{\mathrm{ph}}$, which are contributed by electron (mobile carrier) and phonon. Since $\kappa_{\mathrm{el}}$ can be controlled by the modification of the electronic transport properties, as described by the Wiedemann–Franz law [3], finding a material with essentially low $\kappa_{\mathrm{ph}}$ with a large $S^2\sigma$ has been one of the strategies for developing high-$ZT$ materials, typically with $ZT > 1$ [1]. One of the key strategies for improving the TE properties is creating a layered structure. Several layered compounds, such as $Bi_2Te_3$, Co oxides, and $CsBi_4Te_6$, exhibit high thermoelectric performance [4–7]. In layered systems, low-dimensional electronic states, a structure composed of a stacking of sheets, and/or a large unit cell could be advantageous for producing high TE properties. In addition, nano-structuring and band convergence have been found to be effective to improve $ZT$ in TE materials such as PbTe [2,8]. Another way to reduce $\kappa_{\mathrm{ph}}$ is the use of lattice anharmonicity [9–12]. This strategy is relatively new, but various TE materials have been discovered to exhibit a high TE performance [13–15]. Recently, the introduction of high-entropy states in TE materials has been developed [2,16–20]. Due to the introduced disorder, $\kappa_{\mathrm{ph}}$ is expected to be largely suppressed, and a high $ZT\sim2$ is observed in high-entropy alloy-type (HEA-type) chalcogenides [19]. However, a strategy to obtain a low $\kappa_{\mathrm{ph}}$ by entropy control or the effect on anharmonicity has not been established. Therefore, in this article, we addressed this issue on HEA-type TE materials using the Grüneisen parameter ($\gamma_{\mathrm{G}}$) [21], which will be reviewed in the next section.

Other functional materials important for solving the energy problem are the super-conductors (SC). As it is well known, SCs exhibit zero-resistivity states at temperatures lower than their SC transition temperature ($T_{\mathrm{c}}$). For most superconductors, SC states emerge through the formation of electron pairs called Cooper pairs, which is achieved via electron–phonon interactions [22]. Therefore, the mechanisms of superconductivity mediated by phonons for most superconductors are classified as being of the conventional type. Since 1986, high-$T_{\mathrm{c}}$ SCs were discovered in Cu-based [23] and Fe-based [24] SCs, and their SC mechanisms are believed to be unconventional, namely, not mediated by phonons. However, conventional mechanisms have been regarded as a promising way to achieve a high $T_{\mathrm{c}}$ due to the recent development of hydrogen-based SC materials under extremely high pressure [25,26]. In addition, lattice anharmonicity has been considered as a key factor for SC in hydrogen-based materials [27]. Therefore, the understanding of lattice anharmonicity in SCs is also an important issue. As in HEA-type TE materials, the effect of high entropy has recently been introduced in various SC [28–37]. In HEA-type SCs, the effect of disorder on electronic and phonon characteristics and the modification of anharmonicity have not been addressed yet. Therefore, knowledge about the relationship between high-entropy states and anharmonicity would be useful for further design of SCs with a high $T_{\mathrm{c}}$.

Motivated by this background, we studied the anharmonicity of two-dimensional and three-dimensional systems with different configurational entropy of mixing, from zero entropy to HEA states. To investigate the evolution of anharmonicity with the increase in configurational entropy, we used $\gamma_{\mathrm{G}}$ in this study. As a conclusion, we propose that anharmonicity in both two- and three-dimensional structures can be modified by the increase in configurational entropy. The suppression of anharmonicity in HEA states would be a common feature in HEA-type materials with different structural dimensionalities.

*1.2. Grüneisen Parameter ($\gamma_{\mathrm{G}}$)*

The $\gamma_{\mathrm{G}}$ of inorganic materials is calculated using the following formula; $\gamma_{\mathrm{G}} = \beta_{\mathrm{V}} B V_{\mathrm{mol}}/C_{\mathrm{V}}$, where $\beta_{\mathrm{V}}$, $B$, $V_{\mathrm{mol}}$, and $C_{\mathrm{V}}$ are the volume thermal expansion coefficient, the bulk modulus, the molar volume, and the specific heat, respectively. The parameters needed for the estimation of $\gamma_{\mathrm{G}}$ are calculated as follows

$$\beta_{\mathrm{V}} = \frac{1}{V(300\ \mathrm{K})}\frac{dV}{dT} \tag{2}$$

$$B = \rho\left(v_{\mathrm{L}}^2 - \frac{4}{3}v_{\mathrm{s}}^2\right) \tag{3}$$

$$\theta_{\mathrm{D}} = \left(\frac{h}{k_{\mathrm{B}}}\right)\left[\frac{3n}{4\pi}\left(\frac{N_{\mathrm{A}}\rho}{M}\right)\right]^{\frac{1}{3}}v_{\mathrm{m}} \tag{4}$$

$$v_{\mathrm{m}} = \left[\frac{1}{3}\left(\frac{2}{v_{\mathrm{s}}^3} + \frac{1}{v_{\mathrm{L}}^3}\right)\right]^{-\frac{1}{3}} \tag{5}$$

In the formulas, $dV/dT$, $\rho$, $v_{\mathrm{L}}$, $v_{\mathrm{s}}$, $v_{\mathrm{m}}$, $\theta_{\mathrm{D}}$, $h$, $k_{\mathrm{B}}$, $n$, $N_{\mathrm{A}}$, and $M$ denote the temperature gradient of the lattice volume, the density of the material, the longitudinal sound velocity, the shear sound velocity, the average sound velocity, the Debye temperature, the Plank's constant, the Boltzmann's constant, the number of atoms in the molecule (formula unit), the Avogadro's constant, and the molecular weight (per formula unit). Although the absolute value of $\gamma_{\mathrm{G}}$ of functional materials depends on their crystal structure or constituent elements, at least, the value becomes a good index for discussing the evolution of lattice anharmonicity by doping, pressurizing, or element substitution in similar compounds.

For example, in the metal telluride system $Pb_{1-x}Sn_xTe$, which is a TE material family [8] and a parent phase of SCs with topological electronic states [38,39], the solution of Sn and Pb results in an increase of $\gamma_{\mathrm{G}}$: $\gamma_{\mathrm{G}} = 1.5$, 2.5, 2.8, and 2.1 for $x = 0$, 0.25, 0.5, and 1 [40]. By the enhanced lattice anharmonicity in the alloy phase ($x = 0.25$ and 0.5), $\kappa_{\mathrm{ph}}$ is clearly suppressed, and the effect was explained by the changes in $\gamma_{\mathrm{G}}$ [40]. In our previous study on $LaOBiS_{2-x}Se_x$, which is a TE system [15] and parent phase of layered SCs [41], we revealed that the anharmonic lattice vibration is the origin of a low $\kappa_{\mathrm{ph}}$, using neutron inelastic scattering [12]. By partial Se substitution in $LaOBiS_{2-x}Se_x$, phonon softening was observed, and $\kappa_{\mathrm{ph}}$ decreased with the decrease of the low-energy phonon energy. This trend was reproduced by the Se concentration dependence of $\gamma_{\mathrm{G}}$ in $LaOBiS_{2-x}Se_x$ [42]. As described above, estimation of $\gamma_{\mathrm{G}}$ is a good way to investigate the evolution of lattice anharmonicity in functional materials. Therefore, in this article, we investigated $\gamma_{\mathrm{G}}$ for HEA-type compounds, because these materials, including TE and SC materials, have recently been drawing attention as electronic materials. Here, we estimated $\gamma_{\mathrm{G}}$ for the layered $BiS_2$-based system $REO_{0.7}F_{0.3}BiS_2$ with an HEA-type RE (rare-earth) site and for Mte with an HEA-type M (metal) site.

### 1.3. Motivation of the Study

In our previous work on RE(O,F)BiS$_2$, we synthesized polycrystalline samples of $REO_{0.5}F_{0.5}BiS_2$ with different configurational entropy of mixing ($\Delta S_{\mathrm{mix}}$) at the RE site [28], according to a compositional guideline established for alloy-based HEAs [43,44], where the HEA composition is defined as one containing five or more elements with a composition range of 5–35%; $\Delta S_{\mathrm{mix}}$ is defined as $\Delta S_{\mathrm{mix}} = -R\sum_i c_i \ln c_i$, where $c_i$ and $R$ are the compositional ratio and the gas constant, respectively [44]. Some of the alloy-based HEAs exhibit superconductivity, and the observed superconducting characteristics of HEAs were clearly different from those of pure or low-entropy alloys and amorphous [45–56]. Therefore, the investigation on the effects on local structure and superconducting properties by the introduction of high configurational entropy should be important for understanding superconductivity in HEA-type materials. In the layered compound RE(O,F)BiS$_2$, by changing the number of element and composition at the RE site, samples with different $\Delta S_{\mathrm{mix}}$ were systematically synthesized [56]. Notably, the SC shielding fraction was improved via the suppression of in-plane Bi-S1 local disorder (local distortion) (see Figure 1e for crystal structure and the definition of the S1 site) by an increase in the $\Delta S_{\mathrm{mix}}$ in $REO_{0.5}F_{0.5}BiS_2$ [56]. Since all the examined samples had similar lattice constants, we concluded that the increase in $\Delta S_{\mathrm{mix}}$ resulted in the modification of the local structure and of the SC properties of

$REO_{0.5}F_{0.5}BiS_2$ [56]. Therefore, the estimation of $\gamma_G$ for RE(O,F)BiS$_2$ would be useful to understand the effect explained above. In addition, we investigated the effect of $\Delta S_{mix}$ on $\gamma_G$ of metal telluride (Mte) to understand the generalizability of the effects.

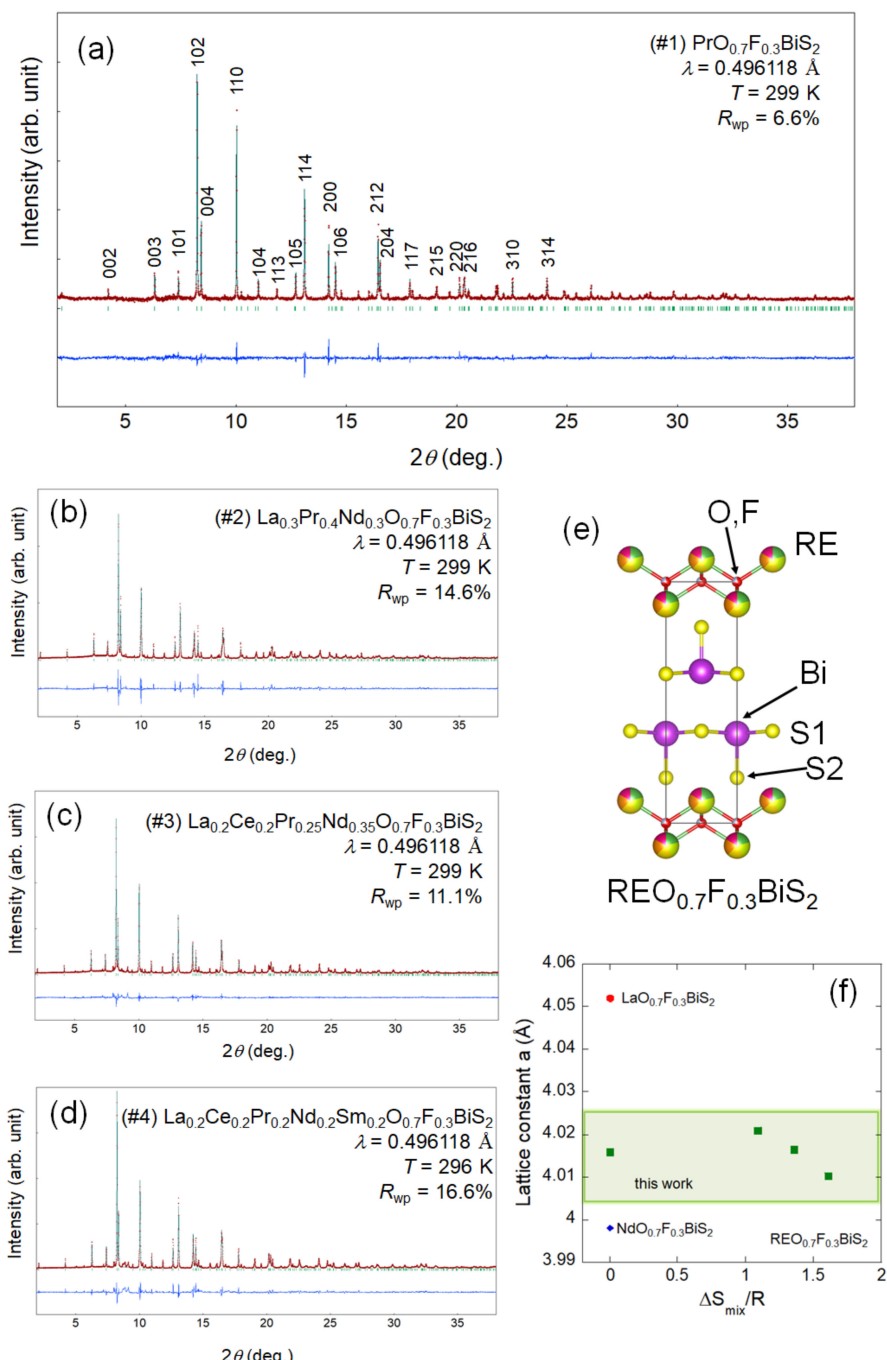

**Figure 1.** SXRD patterns and Rietveld refinement results for (**a**) #1, (**b**) #2, (**c**) #3, and (**d**) #4. The numbers in Figure 1a are Miller's indices. (**e**) Schematic image of the crystal structure of $REO_{0.7}F_{0.3}BiS_2$. (**f**) $\Delta S_{mix}$ dependence of the lattice constant a at $T = 300$ K for the examined samples and RE = La, Nd [41].

## 2. Results

In this study, we designed and synthesized polycrystalline samples of $REO_{0.7}F_{0.3}BiS_2$ according to the material design concept described in [45]. We fixed F concentration to a uniform electron-doping level; the composition of $REO_{0.7}F_{0.3}BiS_2$ was that typical of the superconducting phase of the BiS$_2$-based family. Considering the in-plane chemical

pressure, which was tuned by the *a*-axis in the BiS$_2$-based system [41], we chose the RE site composition to prepare samples with similar constant *a* after trial and error. As shown in Figure 1f, the lattice constant *a* for all samples was similar. Information for the REO$_{0.7}$F$_{0.3}$BiS$_2$ samples (sample #1–#4) is reported in Table 1. By changing the RE-site elements, $\Delta S_{mix}$ was systematically tuned, and sample #4 could be regarded as an HEA-type compound because of $\Delta S_{mix} > 1.5\,R$. The relative density and the values of $v_L$ were measured using high-pressure annealed samples. We found that $v_L$ slightly decreased with the increase of $\Delta S_{mix}$. To evaluate $\beta_V$, the temperature dependence of the lattice volume was measured using synchrotron XRD (SXRD). Figure 1 shows the Rietveld refinement results of the SXRD patterns measured at temperatures near 300 K. Although small impurity peaks of RE$_2$O$_2$S and/or REF$_3$ were found, the single-phase analysis appeared sufficient for the estimation of the lattice volume using the Rietveld refinement. The temperature dependence of the estimated lattice volume (*V*) was determined by refining all the SXRD patterns collected at different temperatures and plotted in Figure 2. V linearly increased with the increasing temperature. By linear fitting of the data and Formula (2), $\beta_V$ was estimated and is presented in Table 1.

**Table 1.** Sample information, including nominal composition, $\Delta S_{mix}/R$, relative density, sound velocity ($v_L$ and $v_S$), volume thermal expansion coefficient ($\beta_V$), Debye temperature ($\theta_D$), bulk modulus (*B*), and $\gamma_G$.

| Sample No. | #1 | #2 | #3 | #4 |
|---|---|---|---|---|
| RE site | Pr | La$_{0.3}$Pr$_{0.4}$Nd$_{0.3}$ | La$_{0.2}$Ce$_{0.2}$Pr$_{0.25}$Nd$_{0.35}$ | La$_{0.2}$Ce$_{0.2}$Pr$_{0.2}$Nd$_{0.2}$Sm$_{0.2}$ |
| $\Delta S_{mix}/R$ (RE) | 0 | 1.09 | 1.36 | 1.61 |
| Relative density | 99% | 97% | 97% | 99% |
| $V_L$ (m/s) | 3430 | 3400 | 3320 | 3260 |
| $V_S$ (m/s) | 1860 | 1730 | 1720 | 1850 |
| $\beta_V$ (1/K) | 0.0000369 | 0.0000356 | 0.0000365 | 0.0000388 |
| $\theta_D$ (K) | 221 | 207 | 205 | 219 |
| $B$ (GPa) | 47.9 | 54.4 | 50.9 | 41.8 |
| $\gamma_G$ | 0.94 | 1.02 | 0.98 | 0.86 |

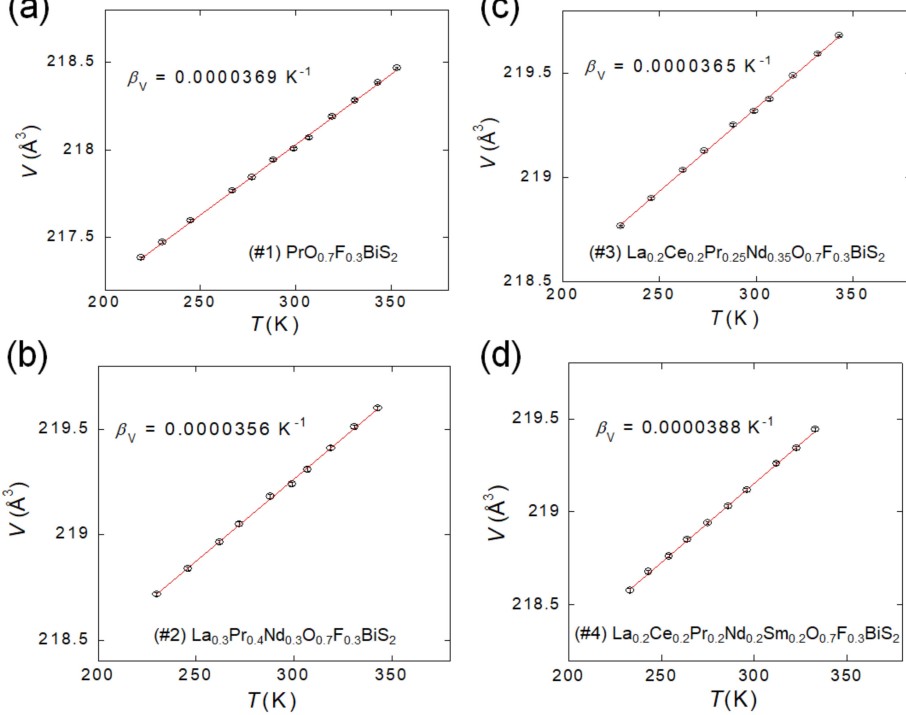

**Figure 2.** Temperature (*T*) dependence of the lattice volume (*V*) for (**a**) #1, (**b**) #2, (**c**) #3, and (**d**) #4. The red lines show the linear fitting results; the estimated $\beta_V$ is displayed.

Figure 3 shows the results of specific heat ($C$) for samples #1–#4. The data were analyzed based on a model for the low-temperature region; $C = \gamma T + \beta T^3$, where $\gamma$ and $\beta$ are specific heat constants of electronic and phonon contributions, respectively. $\theta_D$ was calculated from $\beta$ using the following formula

$$\beta = \frac{12\pi^4 N_A k_B}{5\theta_D^3} \tag{6}$$

where $C_V$ was calculated using the Dulong–Petit law, which gave $C_V = 3R$. $V_{mol}$ was calculated using ideal density and molar mass estimated from the compositions. The parameters needed for calculating $\gamma_G$ for $REO_{0.7}F_{0.3}BiS_2$ are listed in Table 1. The estimated $\gamma_G$ is plotted in Figure 4a. The $\gamma_G$ of #2 was larger than that of #1. The trend of $\gamma_G$ increasing with a small increase of $\Delta S_{mix}$ is similar to the trend reported for $Pb_{1-x}Sn_xTe$ [40]. However, with a further increase of $\Delta S_{mix}$, $\gamma_G$ decreased in the middle-to-high entropy region. These results would suggest that the anharmonicity in $REO_{0.7}F_{0.3}BiS_2$ was enhanced in the low-entropy region and was suppressed in the middle- and HEA regions. To explore the commonality of this trend, we plotted the data presented in [40] in Figure 4b and added $\gamma_G$ for HEA-type metal telluride ($AgInSnPbBiTe_5$) in the plot.

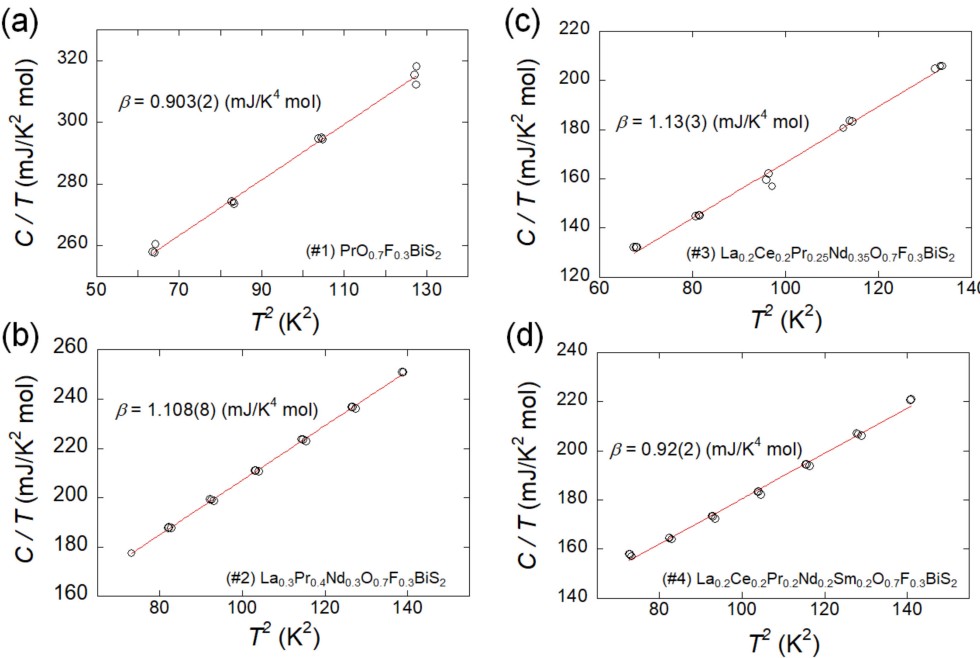

**Figure 3.** $T^2$ dependence of specific heat ($C/T$) for (**a**) #1, (**b**) #2, (**c**) #3, and (**d**) #4. The red lines show linear fitting results; the estimated phonon-contribution constant ($\beta$) is displayed.

The temperature dependence of SXRD and the low-temperature specific heat were measured for $AgInSnPbBiTe_5$. Figure 5a,b shows the results of Rietveld refinement of the SXRD pattern and the temperature evolution of $V$. We estimated $\beta_V = 0.0000672 \ K^{-1}$. Figure 5c shows the low-temperature specific heat and the fitting result using the formula $C = \beta_1 T^3 + \beta_2 T^5$, which was used in previous work on MTe [57]. Using $\beta_1$, $\theta_D$ was calculated as 136 K. The $v_L$ obtained at high density (relative density of ~100%) was 2740 m/s. The calculated $B$ was 33.8 GPa, and the obtained $\gamma_G$ was 1.94. The trend of $\gamma_G$ calculated for MTe (Figure 4b) is quite interesting. As highlighted by the green line, $\gamma_G$ of MTe decreased in the HEA region; this trend is similar to that shown in Figure 4a ($REO_{0.7}F_{0.3}BiS_2$). We considered that this trend would be caused by entropy tuning and that this is possibly a universal feature for HEA-type functional materials. In the next section, we will briefly discuss the possible origin of the $\Delta S_{mix}$ dependence of $\gamma_G$.

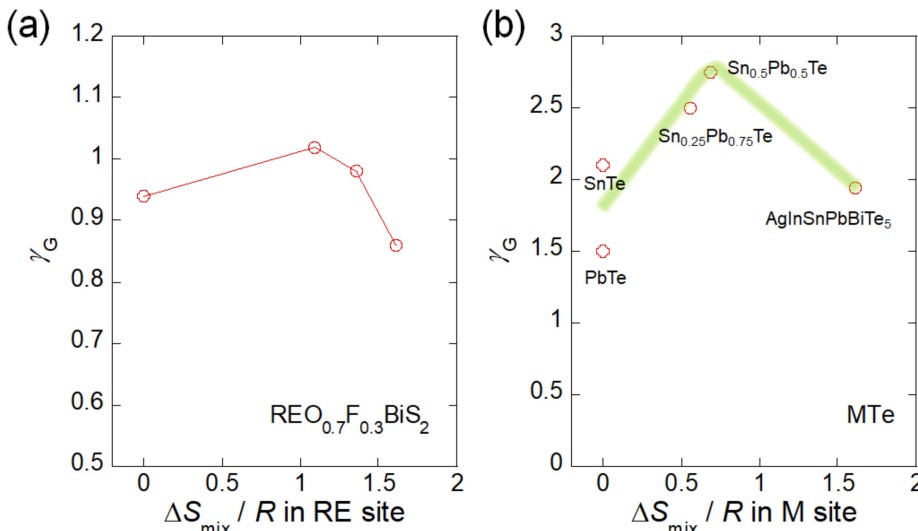

**Figure 4.** Estimated $\gamma_G$ for (**a**) $REO_{0.7}F_{0.3}BiS_2$ and (**b**) MTe plotted as a function of $\Delta S_{mix}/R$. The data for $Pb_{1-x}Sn_xTe$ were taken from [39]. The green line is an eye guide.

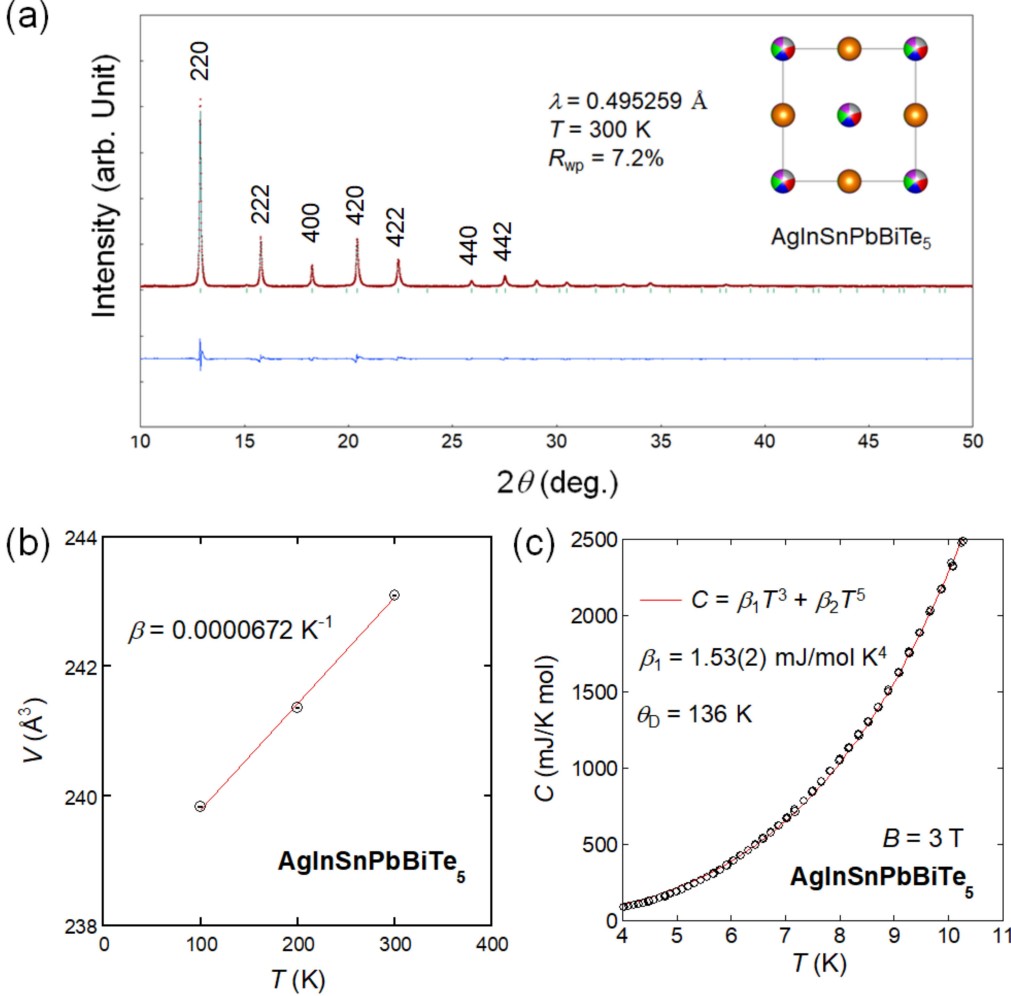

**Figure 5.** (**a**) SXRD pattern and Rietveld refinement results for $AgInSnPbBiTe_5$. The numbers in Figure 1a are Miller's indices. The inset shows a schematic image of the crystal structure. (**b**) Temperature dependence of $V$ for $AgInSnPbBiTe_5$. (**c**) Specific heat data for $AgInSnPbBiTe_5$ plotted as a function of $T$. The red line shows the fitting results.

## 3. Discussion

Entropy is a general concept related to various physical quantities. To discuss the phenomena observed in this work, we have to consider at least two different entropies, i.e., the above-mentioned configurational entropy of mixing and the vibrational entropy. In a field of glass transition, configurational and vibrational entropies have been separately considered and analyzed, and their interplay has been discussed [58–60]. On the basis of such a concept established for metallic glasses, we will discuss the origin of the results shown in Figure 4. In the low-to-middle-entropy region, the small disorder introduced by element substitution affected the average atomic positions, and weak displacements of the atoms and/or bonds were introduced. As revealed by the extended X-ray absorption fine structure of $Pb_{1-x}Sn_xTe$ [40], bond anharmonicity was enhanced in the alloyed region, resulting in a large $\gamma_G$. Basically, structural disorder should be enhanced by an increase in $\Delta S_{mix}$ for both $REO_{0.7}F_{0.3}BiS_2$ and MTe systems. Therefore, in the HEA region, the local structure (atomic positions and bonds) should approach glass-like states. In such a case, the total entropy in the system would be governed by the configurational entropy, resulting in the suppression of the vibrational entropy. Although the results in this study are not exhibiting direct evidence of the interplay between configurational and vibrational entropies, this concept would be useful to develop functional materials with lattice anharmonicity via entropy tuning. To establish this new concept, further investigation of $\gamma_G$ and lattice anharmonicity of functional materials are necessary. Investigation on materials with harmonic lattice vibration is also needed.

## 4. Materials and Methods

The polycrystalline samples of $REO_{0.7}F_{0.3}BiS_2$ (see Table 1 for the nominal composition of the examined four samples indicated as #1–#4) were synthesized by a solid-state reaction method. Powders of $La_2S_3$ (99.9%), $Ce_2S_3$ (99.9%), $Pr_2S_3$ (99.9%), $Nd_2S_3$ (99%), $Sm_2S_3$ (99.9%), $Bi_2O_3$ (99.999%), and $BiF_3$ (99.9%) and grains of Bi (99.999%) and S (99.99%) were used. The $Bi_2S_3$ powders were synthesized by reacting Bi and S in an evacuated quartz tube. The mixture of starting materials with a nominal composition was obtained by mortar and pestle, pelletized, and sealed into an evacuated quartz tube. The samples were heated at 700 °C for 20 h for #1 and at 750 °C for 20 h for #2–#4. The obtained products were ground, pelletized, sealed into an evacuated quartz tube, and heated under the same heating conditions for homogenization. Since dense samples are needed for sound velocity measurements, the obtained $REO_{0.7}F_{0.3}BiS_2$ powders were annealed at 400 °C for 15 min using a cubic–anvil-type high-pressure synthesis system under 1.5 GPa. The obtained samples had a relative density higher than 97%. For the $AgInSnPbBiTe_5$ sample, a precursor powder was synthesized by reacting Ag (99.9%) powder and In (99.99%), Sn (99.999%), Pb (99.9%), Bi, and Te (99.999%) grains with a nominal composition at 800 °C in an evacuated quartz tube. The obtained precursor was annealed at 500 °C for 30 min under a high pressure of 3 GPa.

The synchrotron XRD (SXRD) was performed with the beamline BL02B02, SPring-8 (under proposals Nos.: 2020A0068 and 2021B1175). The wavelength of the X-ray was 0.496118(1) Å for the experiments with $REO_{0.7}F_{0.3}BiS_2$ and 0.495259(1) Å for that with $AgInSnPbBiTe_5$. The SXRD experiments were performed with a sample rotator system; the diffraction data were collected using a high-resolution one-dimensional semiconductor detector (multiple MYTHEN system [61]) with a step size of $2\theta = 0.006°$. The temperature of the samples was changed by a $N_2$-gas temperature controller.

The crystal structure parameters were refined using the Rietveld method with the RIETAN-FP program [62]. The tetragonal $P4/nmm$ (#129) model was used for the refinements for $REO_{0.7}F_{0.3}BiS_2$. For $AgInSnPbBiTe_5$, the NaCl-type (cubic $Fm$-$3m$; #225) model was used for the refinements. For the structural parameters, we reported them in earlier works [28,29,41]. The schematic images of the crystal structure were drawn using VESTA software [63].

The temperature dependence of specific heat was measured with a Physical Property Measurement System (PPMS, Quantum Design) by a relaxation method. The longitudinal sound velocity ($v_L$) of the sample was measured on dense samples using an ultrasonic thickness detector (Satotech-2000C). The $v_L$ was corrected using the relative density of the polycrystalline sample [42], and the corrected $v_L$ was used for the calculation of $\gamma_G$.

## 5. Conclusions

In the introduction, the characteristics of TE and SC materials and the importance of lattice anharmonicity in those materials were reviewed. Motivated by recent works on the analysis of anharmonicity and $\gamma_G$, we investigated the structural and physical properties of $REO_{0.7}F_{0.3}BiS_2$, which is a layered (two-dimensional) system, and of the NaCl-type (three-dimensional) metal telluride (MTe); for both systems, $\Delta S_{mix}$ was controlled by changing the solution elements at the RE and M sites. By plotting the estimated $\Delta S_{mix}$, we found that an increase in $\Delta S_{mix}$ in the low-to-middle-entropy region resulted in the enhancement of anharmonicity, but a further increase in $\Delta S_{mix}$ in the middle-to-high-entropy region clearly suppressed anharmonicity. Since this trend was observed in both cases with two- and three-dimensional structures, we propose that this trend would be a universal feature of functional materials in which the configurational entropy of mixing is modified by alloying one or more sites. Further studies on $\gamma_G$ for various functional materials are desired to confirm this concept and will open a new pathway for material development by entropy tuning.

**Author Contributions:** Conceptualization, A.Y. and Y.M.; methodology, F.I.A., Y.N., K.T., R.M., C.M. and Y.M.; validation, A.Y., Y.G. and Y.M.; formal analysis, F.I.A., Y.N., A.Y., M.R.K., M.Y., Y.G., A.M., K.T., R.M., Y.T., C.M. and Y.M.; investigation, F.I.A., Y.N., A.Y., M.R.K., M.Y., Y.G., A.M., K.T., R.M., Y.T., C.M. and Y.M.; resources, Y.G., K.T., R.M., Y.T., C.M. and Y.M.; data curation, F.I.A., A.Y., M.R.K., K.T., R.M., C.M. and Y.M.; writing—original draft preparation, F.I.A. and Y.M.; writing—review and editing, F.I.A., Y.N., A.Y., M.R.K., M.Y., Y.G., A.M., K.T., R.M., Y.T., C.M. and Y.M.; visualization, F.I.A. and Y.M.; supervision, A.Y., Y.G., Y.T. and Y.M.; project administration, Y.G., K.T., C.M. and Y.M.; funding acquisition, Y.G. and Y.M. All authors have read and agreed to the published version of the manuscript.

**Funding:** This work was partially funded by Grant-in-Aid for Scientific Research (KAKENHI) (Nos. 18KK0076, 21K18834, 21H00151), JST-CREST (No. JPMJCR20Q4) and Tokyo Metropolitan Government Advanced Research (No. H31-1).

**Institutional Review Board Statement:** Not applicable.

**Informed Consent Statement:** Not applicable.

**Data Availability Statement:** The data reported in this article can be provided by the corresponding author (Yoshikazu Mizuguchi) upon reasonable request.

**Acknowledgments:** The authors thank R. Kurita, M. Omprakash, and O. Miura for their supports in the experiments and for fruitful discussions.

**Conflicts of Interest:** The authors declare no conflict of interest.

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
