# Peer review of "Estimation of the Grüneisen Parameter of High-Entropy Alloy-Type Functional Materials: The Cases of REO0.7F0.3BiS2 and MTe"

_condensedmatter, doi:10.3390/condmat7020034_

Round 1

Reviewer 1 Report

This is an interesting article that studies Grüneisen parameter for recently-developed high-entropy-alloy-type functional materials with a layered and a NaCl-type structures. Four samples of the layered (RE)O0.7F0.3BiS2 compounds with RE = Pr, La0.3Pr0.4Nd0.3, La0.2Ce0.2Pr0.25Nd0.35, La0.2Ce0.2Pr0.2Nd0.2Sm0.2, are investigated in this work. In high-entropy-alloys and superconductors, the anharmonicity in structures is considered by the authors as the way to increase in configurational entropy, and go over zero entropy and HEA states.

The manuscript contains a broad review of the thermoelectric materials and superconductors, as well as Grüneisen parameter in different types of the thermoelectric and superconducting materials with a large number of relevant references. Pages 6-7 also report the AgInSnPbBiTe5 sample study. The temperature dependence of lattice volume and specific heat for the Grüneisen parameter estimations are reported.

Although the manuscript contains novel results and some indirect evidence of the interplay between configurational and vibrational entropies, the title of the manuscript is too general. Please include at least the compounds (RE)O0.7F0.3BiS2 and AgInSnPbBiTe5 to the title.

The motivation and plan of the study are clear, however, the reader can find the lack of enough evidence for the proposed concept because of a few systems studied.

After these minor changes the manuscript can be accepted to Condensed Matter. 

Author Response

Thank you so much for reviewing the manuscript. Please see the attached.

Reviewer 2 Report

The manuscript reports: Estimation of Grüneisen Parameter of High-entropy-alloy-type Functional Materials.

The authors should consider some points before accepting the manuscript in this journal.

The introduction of relevant background and research progress was not comprehensive enough. Regarding the introduction, the motivation is absent, and some sentences must be referenced.

The Grüneisen parameter (γ G), must be referenced. The first paragraph of the results is a summary of previous work, my suggestion is to use this in the introduction as part of the motivation of the presented work.

In this presented work, the most relevant results that the authors consider important for publication must be shown. Under what criteria the dopants and the percentage entered are selected (Table 1)?

About figure 1. First, the form as figure was presented does not allow to see the most relevant details of the refinement, which I suggest modifying the format of the figure, so that all details can be observed, since these refinements are the basis of the next discussions.

Miller's indices for each crystallographic plane must be entered.

Through refinement, several structural parameters can be obtained and have not been shown here. For example, lattice constant? lattice strain? How much did the dopants change the angles in the crystal structure?

In addition, Figure 1 is only mentioned, but the information presented was not discussed.

Include the error bar for the parameters shown in figure 3 and figure 4. Miller indices should be included in figure 5a. The crystallographic card for each analyzed system must be shown.

The quality of all figures should be improved, some information within them has small font sizes, making it difficult to observe.

It is appreciable that the authors analyzed the Estimation of Grüneisen Parameter of High-entropy-alloy-type Functional Materials, but there is no in-depth scientific discussion.

The text is very descriptive and qualitative. How many times were the experiments repeated to reach the conclusions shown? What is the reproducibility of this study? A careful review of the entire manuscript must be performed.

Strong scientific discussions must be made throughout the entire manuscript and the results must be compared with those reported in the literature. To strengthen the discussions, more up-to-date bibliographic references must be sought and included in the text.

Author Response

(The authors gave the same response as above.)

Round 2

Reviewer 2 Report

The authors have reviewed the manuscript and this version can be accepted for publication